# Modernizing Total Hip Arthroplasty Perioperative Pathways: The Implementation of ERAS-Outpatient Protocol

**DOI:** 10.3390/jcm11123293

**Published:** 2022-06-08

**Authors:** Sivan Sivaloganathan, William G. Blakeney, Pascal-André Vendittoli

**Affiliations:** 1Surgery Department, Hôpital Maisonneuve-Rosemont, Montreal University, 5415 Boulevard de l’Assomption, Montreal, QC H1T 2M4, Canada; sivan_shankar@hotmail.co.uk; 2Department of Orthopedic Surgery, Royal Perth Hospital, Perth, WA 6000, Australia; blakeney@gmail.com; 3Clinique Orthopédique Duval, Laval, QC H7M 2Y3, Canada; 4Personalized Arthroplasty Society, Atlanta, GA 30305, USA

**Keywords:** arthroplasty, knee, hip, enhanced recovery after surgery, fast-track, outpatient, patient outcome, complication

## Abstract

Pressure to reduce healthcare costs, limited hospital availability along with improvements in surgical technique and perioperative care motivated many centers to focus on outpatient pathway implementation. However, in many short-stay protocols, the focus has shifted away from aiming to reduce complications and improved rehabilitation, to using length of stay as the main factor of success. To improve patient outcomes and maintain safety, the best way to implement a successful outpatient program would be to combine it with the principles of enhanced recovery after surgery (ERAS), and to improve patient recovery to a level where the patient is able to leave the hospital sooner. This article delivers a case for modernizing total hip arthroplasty perioperative pathways by implementing ERAS-outpatient protocols.

## 1. Introduction

The Enhanced Recovery After Surgery (ERAS) principles, proposed by Dr Husted and Dr Kehlet in 1997, aimed to reduce the duration of a patient’s hospital stay by delivering optimized patient-centered care, and ultimately, to achieve a “pain and risk-free operation” [1,2]. The core aspects of the ERAS protocol incorporate preoperative patient education and medical optimization. In addition, ERAS aims at perioperative improved pain control, blood conservation, nutrition support, early mobilization, maintenance of gastrointestinal function, and the reduction of adverse events [3].

The COVID-19 pandemic has impacted healthcare systems across the globe. Over the course of the pandemic, healthcare teams have adjusted their practice, which has enabled non-essential procedures to resume and the sizeable backlog of procedures to be addressed. One appealing solution is to implement outpatient protocols for patients undergoing hip arthroplasty [4,5]. On the other hand, some short-stay protocols have shifted from aiming to reduce complications and improve recovery to using length-of-stay as the primary measure of success. Combining the principles of ERAS as part of an innovative integrated ERAS-outpatient protocol, hospitals have the opportunity to reduce the challenges of bed capacity and reduce healthcare costs whilst optimizing the patient experience, clinical safety, and clinical effectiveness [6,7].

Achieving patient discharge on the day of surgery relies on adequate patient selection and identification of the factors causing outpatient failures, and their prevention. Although patient selection should not be too restrictive, the identification of some important contra-indications is important for patients’ safety (see Figure 1). Factors known to prevent discharge on the day of surgery include: pain, nausea, dizziness, orthostatic hypotension, wound discharge, urinary retention, and loss of muscle function. The occurrence of these adverse events is not universal, and it is difficult to predict to whom they will happen. We believe they can be prevented systematically through a multimodal approach using the ERAS principles. All patients should be managed by applying five core principles that emerge from ERAS: patient and family engagement; surgical best practices; multimodal opioid-sparing pain management; mobility and physical activity; and fluid and nutrition management [7].

## 2. Patient and Family Engagement

The impetus of preoperative education is to improve patients’ knowledge, manage their expectations, and have a positive impact on their lifestyle choices. Before surgery, patients are encouraged to optimize modifiable risk factors, focusing on improving nutrition, as well as the cessation of smoking and alcohol. There is also early involvement of a physiotherapist to promote preoperative exercises [8]. 

Clinicians are advised to discuss pain management with patients to help guide treatment decisions and shape realistic expectations regarding pain at the preoperative stage. Patients’ drugs and doses should be documented before surgery; this identifies opioid-tolerant/reliant patients and affords time to manage them appropriately. Patients are also screened for anxiety and depression, to identify those who may require additional counselling/psychological services to prevent symptoms from worsening. In addition to the clinical aspects, logistical issues such as patient transport and home care should be addressed before the surgery date. 

A summary of all the recommendations should be available in a patient roadmap booklet and/or through an online platform. The day of surgery can be a stressful experience; therefore, patients should receive a detailed overview in advance [9].

## 3. Patient Optimization

It is well known that specific conditions are linked to an increased rate of postoperative complications. For example, a study by Evans et al. demonstrated that a preoperative hemoglobin level below or equal to 120 g/L was associated with an incidence of blood transfusion of 47.7% versus 7.2% when above this threshold. [10]. Similarly, it is evident from the literature that poorly controlled diabetes (defined as HbA1c of >8%) is associated with a higher incidence of wound complications and infections [11,12]. A study by Han et al. demonstrated that poor preoperative glycemic control is associated with a greater risk of postoperative wound complications in lower-limb arthroplasty patients [13].

The preoperative period provides a unique opportunity to optimize all modifiable risk factors. Both nursing and medical evaluations should identify and address patient habits (such as smoking and excessive alcohol consumption) and comorbidities, such as ischemic heart disease, high blood pressure, sleep apnea, and malnourishment [14,15,16]. 

Men with prostatitis symptoms should be identified pre-operatively using the 7-question International Prostate Symptoms Score (IPSS), with the aim of preventing postoperative urinary retention. Patients at risk of urinary retention secondary to prostatitis should be prescribed oral tamsulosin, which should commence three nights before hospital admission. Venous insufficiency with chronic lower-limb swelling should also be treated with compressive stockings pre- and post-operatively.

Non-modifiable conditions—for example, renal insufficiency—which may limit NSAID use in the ERAS protocol should be factored in to avoid complications. Drugs prescribed to treat pre-existing conditions should be continued before the surgery. Discontinuation and resumption of these drugs should be determined by a person with specialized knowledge of these conditions. Patients should be advised not to receive intra-articular injections in the 3–6 months leading to surgery [8,17,18]. 

## 4. Infection Prevention

Optimizing patient risk factors such as tobacco smoking, excessive alcohol consumption, diabetic control, and anemia will contribute to reducing the risk of postoperative infection. Patients should apply a 4% aqueous chlorhexidine gluconate wash on the morning of surgery. Shaving at the surgical site with a razor should be avoided, as micro-abrasion on the skin provides areas for bacteria to remain and replicate in the surgical field [19,20,21]. Antibiotic prophylaxis with a first- or second-generation cephalosporin (i.e., cefazolin or cefuroxime), administered intravenously 30 to 60 min before incision as a single and weight-adjusted dose, is recommended. Cephalosporin is more effective than vancomycin as a prophylactic agent for patients undergoing orthopedic procedures. If treatment with vancomycin is necessary, administering it in combination with other antibiotics should be considered to increase therapeutic efficacy [22,23]. It is well known that post-operative allogenic blood transfusion increases the risk of deep infection, hence the importance of effective blood conservation management [24]. 

## 5. Blood Conservation

Total hip arthroplasty (THA) surgery poses a risk of substantial perioperative blood loss. This risk can be minimized to avert the need for transfusion in a number of ways, including preoperative hemoglobin optimization, perioperative systemic tranexamic acid administration, careful surgical hemostasis, and the admission of adrenalin into the local anesthetic infiltration [10,25,26,27]. 

Maintaining a higher post-operative hemoglobin level will improve patient recovery and reduce the risk of ischemic cardiac events and the need for transfusion. It is of primary importance to avoid severe anemia when patients are discharged promptly after surgery, as outpatient monitoring of hemoglobin levels is complex [28,29]. 

## 6. Pain Control

Improved pain control is one of the main pillars of patient satisfaction after surgery [30]. The management of postoperative pain has broad ramifications; some discomfort post-surgery is unavoidable, and depending on the severity, it can affect functional recovery, reduce patient satisfaction, and prolong the duration of the hospital stay. Postoperative pain control has traditionally been managed with opioids, known to produce adverse effects such as drowsiness, nausea, vomiting, constipation, and respiratory depression. A multimodal analgesic approach should be implemented to minimize or avoid opioid use to promote recovery. 

The ERAS multimodal approach to pain control includes patient engagement and education, a multimodal pre-emptive medication regime which may include: acetaminophen, NSAIDs, dexamethasone, a long-acting opioid, and pregabalin. In the operating room, a combined epidural (xylocaine)–sedation(propofol) or a short acting spinal anesthesia with chloroprocaine associated with periarticular tissue infiltration with large doses of local anesthetics have been shown to be effective in minimizing post-operative pain. After surgery, a multimodal approach with oral drugs (acetaminophen, NSAIDs, tramadol, and opioids as a last resort), combined with local cryotherapy, should be used [7,8,31,32,33,34,35]. Using this multimodal approach, the need for postoperative opioids can be avoided in most patients.

## 7. Early Mobilization

With a focus of ERAS being the reduction of hospital stays, early postoperative function and mobilization are essential components of an outpatient program. The impact of the anesthetic approach on perioperative outcomes is controversial. The ERAS anesthesia technique for hip arthroplasty should aim to reduce postoperative pain, minimize the risk of orthostatic hypotension or urinary retention, and limit motor function loss during and after surgery. Avoidance of perioperative opioids and benzodiazepines with an anesthetic technique that minimizes the loss of lower-limb motor function and the risk of postoperative orthostatic hypotension is ideal. Moreover, catheterization and wound drains should be avoided to aid in early mobilization. The use of preoperative dexamethasone (6–10 mg IV) and an opioid-free, combined epidural–sedation technique to avoid a complete motor block can allow the patient to move their feet during the surgical procedure and allow patients to stand and walk as soon as 2–3 h post-surgery. This anesthetic approach can minimize altered autonomic function, orthostatic hypotension, and urinary retention rates. By adopting this multimodal technique, the regulation of cortisol and adrenocorticotrophic hormone levels is better maintained. Range-of-motion and weight-bearing restrictions should be avoided to facilitate early mobilization [6,36,37,38]. 

## 8. Gastrointestinal Function

Postoperative nausea and vomiting delay eating and mobilization for the patient and, hence, impede recovery [8]. The causes of postoperative nausea are extensive, including the combination of various anesthetic agents, hypovolemia, prolonged perioperative fasting, anemia, immobilization, and opioid use. The ERAS multimodal approach includes a limited-fasting pre-surgery protocol, allowing patients to eat solid food up to 6 h before surgery and clear liquids up to 2 h before surgery. Anti-nausea prophylactic drugs should be given systematically (a combination of aprepitant, dexamethasone, and scopolamine patch), as well as opioid avoidance [7,39,40]. Furthermore, patients are encouraged to eat soon after surgery to promote gastrointestinal function [8,41].

## 9. Prevention of Thromboembolic Events

Major surgery and prolonged immobilization can increase the risk of a thromboembolic event. Most deep vein thrombosis (DVT) events occur in the first 24 h after the start of anesthesia. This adverse event can be mitigated through interventions such as intermittent compression of the lower limbs, early mobilization, and oral anticoagulant therapy. These are critical aspects of ERAS [8,42,43]. In a recent international consensus on venous thrombosis prevention, low-dose aspirin (80–100 mg) was recommended as the safest and most effective method of prophylaxis in patients undergoing hip or knee arthroplasty: “We recommend the use of low-dose acetylsalicylic acid (ASA) as the primary method of venous thromboembolism (VTE) prophylaxis in all patients undergoing THA, including moderate- to high-risk patients” [44].

## 10. Wound Management

Wound closure should aim to prevent wound discharge, spontaneous evacuation of hematomas, and patient concern (perception of hemorrhage); moreover, it should minimize dressing changes, and minimize nursing care. One optimized method of sealing the skin can be accomplished using subcuticular skin closure with barbed sutures (or knotless tissue control devices) and skin glue. This seals the wound and prevents discharge or retrograde contamination. Avoiding staples reduces nursing care time and affords the patient the opportunity to resume bathing sooner [8,45,46]. Optimizing wound closure was perceived as one of the most important improvements in the ERAS protocol [30]. 

## 11. ERAS and Outpatient Pathway Integration for THA

In 2014, a team led by Dr Pascal-André Vendittoli introduced a perioperative short-stay protocol for THA procedures at Maisonneuve-Rosemont Hospital, following the ERAS principles to minimize adverse events. Implementing the ERAS short-stay protocols (Figure 1, flow diagram) required the involvement of a multidisciplinary team (inclusive of patients), who committed to a standardized, evidence-based protocol for the perioperative management of the patient [6].

Several studies have reviewed various ERAS-based short-stay protocols. The studies by Vendittoli et al. indicated that the implementation of the ERAS short-stay protocol improved patient care while reducing direct healthcare costs, and shortening the duration of hospital stays when compared with a cohort of patients who followed a standard care pathway [7,30,47]. The mean hospital length of stay for the ERAS short-stay group decreased by 2.8 days for THAs (0.1 vs. 2.9 d, *p* < 0.001). The mean estimated reduction in direct healthcare costs with the ERAS short-stay protocol was CAD 1489 per THA [7]. A study in the US by Van Horne et al. found that implementing their ERAS pathway reduced patient complication rates (Table 1) while improving patient satisfaction (Table 2), with same-day discharge achieved in 84% of patients [48]. A separate study by Hardy et al. found that the patient-centric fundamentals of their specific ERAS-outpatient program delivered a far superior positive patient experience compared to standard inpatient hip replacement pathways [47]. 

## 12. Discussion

Reducing THA to an outpatient procedure raises concerns about higher complication rates for cost savings and increased bed availability. However, in recent years, multiple investigators looking at the complication rates of numerous outpatient programs in comparison to standard inpatient complication rates have concluded that outpatient surgery is not inferior. With the application of ERAS principles and the optimization of patient recovery to enable same-day discharge, we can obtain superior results compared to the inpatient standard of care [7,30,47]. Implementing the ERAS principles in the outpatient setting provides the opportunity to deliver better clinical outcomes for the patient whilst concomitantly providing cost-saving benefits for the healthcare system.

The ERAS protocol’s success depends on the systematic application of its protocols through a multidisciplinary team. Following clearly defined goals that simplify postoperative care is integral to realizing an effective ERAS protocol. A key reason for patients reporting a better THA experience with the ERAS-outpatient pathway was the optimized recovery. Contrary to standard general anesthesia or spinal anesthesia, the ERAS-outpatient opioid and benzodiazepine-free epidural–sedation combination demonstrated improved postoperative wellbeing and a swifter recovery, whilst eliminating the distress associated with situational awareness during the operation [47]. The expedited ambulation following surgery contributed to reduced complications while improving overall function in the immediate postoperative phase. In addition, early mobilization reduced the overall number of physiotherapy sessions required for functional autonomy [39,47,49,50,51]. 

Qualitative studies have demonstrated that effective pain management requires a holistic approach. Patients can be reluctant to take prescribed analgesia for several reasons, including uncertainty about the medication schedule or about the risk of developing an addiction. Patient empowerment through education has proven to be effective. The pre-discharge education delivered by the ERAS-outpatient program helped patients take ownership of their pain control through a combination of information about opioid-sparing analgesia, previous patient feedback, and discussions on the risk of dependency. This method of self-medication, when applied appropriately, has been shown to be strongly associated with better overall pain relief [52,53,54,55,56]. 

Credence should be given to a positive patient experience, because it has been shown to directly correlate with better patient satisfaction and a higher quality of hospital and home care [57]. Patient experience, along with clinical safety and effectiveness, form the three pillars of quality care [58]. Patients valued care providers who were engaged in helping them to recover, and benefited from the availability of the interdisciplinary team by phone to ease their concerns, which improved the overall provider–patient relationship. In addition, in the ERAS-outpatient post-discharge pathway, a caregiver provided a home visit in the early postoperative days to de-escalate any concerns regarding home care. It was McMurray et al. who stated that the dynamic relationship between patients and care providers is a critical component of the patient experience [59].

The ERAS-outpatient program is not without limitations. Patients must be motivated, engaged and have a good comprehension of the protocol. Regrettably, individuals who have a significant coagulation disorder or systemic disease that may require intensivist input, multiple transfusions, or dialysis are necessarily excluded from the program. Patients with psychiatric illnesses that may cause cognitive impairment and frail elderly patients are unable to be part of the ERAS-outpatient program. The absence of local medical services to a patient’s residence, lack of care provider support, and transport logistics could exclude patients from the ERAS-outpatient program. Other relative contraindications to including patients can include a history of long-term urinary catheter requirement post-surgery [8]. 

Although ERAS might revolutionize recovery after an outpatient procedure, the same ERAS principles should be applied to inpatient care, noting that optimizing patient outcomes should be a goal for all patients. ERAS protocols should be applied systematically and have the patient and family at its core to be effective. In most cases, implementing ERAS-outpatient protocols will involve significant changes to practices. Moreover, ERAS protocols should simplify postoperative care as much as possible. For example, the use of oral anticoagulants instead of injectables avoids the need for education on self-injection. In addition, using skin glue to seal the wound reduces wound discharge, and the frequency of dressing changes enables the patient to shower, eliminates the need for staple removal, and results in fewer superficial infections. 

ERAS protocols should improve patient wellbeing to a level that allows them to return home sooner, or even on the day of surgery (outpatient surgery). In recent years, Enhanced Recovery Canada and the Canadian Patient Safety Institute have supported the development of consensus ERAS pathways for THA and TKA. A group of 20 Canadian experts in the field, including nurses, surgeons, anesthetists, physiotherapists, nutritionists, pharmacists, and internal medicine doctors, as well as patients, produced a consensus pathway and other resources. The documents are available on the Healthcare Excellence website: https://www.healthcareexcellence.ca/en/what-we-do/what-we-do-together/enhanced-recovery-canada/resources-for-orthopaedic-surgeries/ (accessed on 10 January 2022).

## 13. Conclusions

In an evolving world in which healthcare pressures are worsening, an adaptive system is required. The ERAS-outpatient pathway delivers a solution. Applying ERAS principles to a short-stay protocol for THA is pivotal to improving patient outcomes, by reducing adverse events and the duration of hospital stays to less than 24 h. Shorter hospital stays after THA increase bed availability in a financially constrained environment. The ERAS-outpatient pathway delivers a standardized care platform which is a patient-centric program that champions patient satisfaction alongside addressing the demands faced by the healthcare provider. It delivers a win–win solution for all stakeholders. 

## Figures and Tables

**Figure 1 jcm-11-03293-f001:**
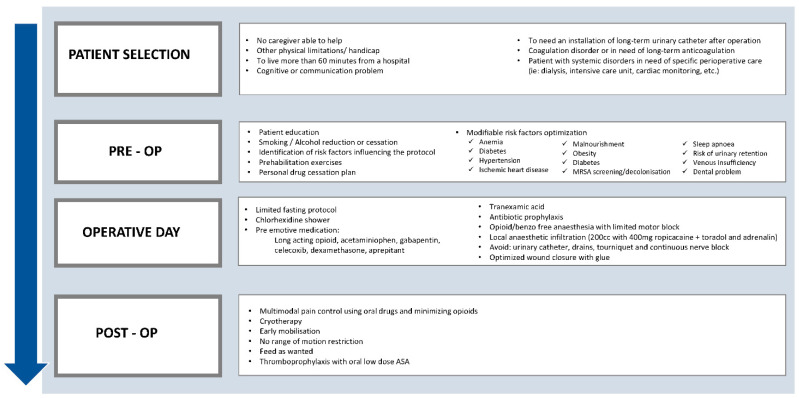
ERAS-outpatient Pathway.

**Table 1 jcm-11-03293-t001:** Complications rates per group (Control vs. ERAS short-stay) according to the Clavien-Dindo classification [7].

Complication	Control (*n* = 150)	ERAS Short-Stay (*n* = 114)	*p*-Value
**GRADE 1**			
Pain	67%	13%	<0.001
Nausea	42%	12%	<0.001
Vomiting	25%	1%	<0.001
Dizziness	15%	4%	0.006
Headache	4%	0%	0.04
Constipation	8%	0%	0.002
Hypotension	26%	11%	0.003
Anemia	8%	0%	0.002
Edema of the operated limb	9%	1%	0.005
Persistent lameness	4%	0%	0.04
Ecchymosis	5%	0%	0.011
Pruritic requiring medication	3%	0%	0.072
Hematoma	3%	0%	0.072
Fall without consequence	3%	5%	0.539
**GRADE 2**			
Urinary retention	13%	4%	0.006
Anemia with transfusion	8%	0%	0.002
Deep vein thrombosis	3%	0%	0.072
Stitch abscess	2%	0%	0.261
Cellulitis	1%	1%	1.0
Sciatic nerve palsy (temporary)	1%	0%	0.507
**GRADE 3**			
Deep infection	0.7%	0.9%	1.0
**GRADE 4**			
Severe respiratory depression	0.7%	0%	1.0
**GRADE 5**			
Death	0%	0%	

**Table 2 jcm-11-03293-t002:** Patients’ Satisfaction on a Visual Analog Scale (0–100) [30]. Comparison of ERAS-outpatient group and the control (standard) group illustrating superior patient satisfaction scores.

Question	ERAS-Outpatient *n* = 47	STD-Inpatient *n* = 47	*p*-Value (Wilcoxon Rank-Signed Test)
How satisfied are you with your time at the hospital (including: anaesthesia, the surgery and physiotherapy)?	96.0 (50.0–100.0, 9.0)	84.6 (0.0–100.0, 23.1)	<0.01
How satisfied are you with pain management post-surgery?	93.2 (18.0–100.0, 13.5)	86.4 (0–100.0, 21.4)	0.002
How satisfied are you with your recovery after surgery? (speed and ease of resuming your usual activities)	95.5 (70.0–100.0, 6.8)	84.4 (1.0–100.0, 19.9)	<0.001
How satisfied are you with your surgical wound closing? method (sutures/glue/staples and bandage)	94.8 (18.0–100.0, 13.2)	85.1 (0.0–100.0, 18.2)	<0.001
How satisfied are you with your surgical experience?	97.1 (75.0–100.0	88.3 (24.0–100.0, 16.2)	<0.001

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
