# Peer review of "Modernizing Total Hip Arthroplasty Perioperative Pathways: The Implementation of ERAS-Outpatient Protocol"

_jcm, 2022, doi:10.3390/jcm11123293_

Round 1

Reviewer 1 Report

The authors provide a review of the Enhanced Recovery After Surgery system and the application for its use today. The content of the article is not new rather is a review of the information that has been available to centers world wide for several years. The merits of the ERAS system have previously been described as well. 

If the authors have data from their institution that shows improvement after the ERAS system was adopted the manuscript would be of more interest. 

Author Response

The authors provide a review of the Enhanced Recovery After Surgery system and the application for its use today. The content of the article is not new rather is a review of the information that has been available to centers world wide for several years. The merits of the ERAS system have previously been described as well. 

If the authors have data from their institution that shows improvement after the ERAS system was adopted the manuscript would be of more interest. 

Though individuals within the Orthopaedic community have attempted to introduce a short stay protocol, the outcomes have been inconsistent. We have included tables of studies performed within our institution. Furthermore, this review article aims to target the wider medical community (General Practitioners) to inform/promote the existence of short stay hip arthroplasty protocols.

Table 1 + Table 2 included. Comparison of ERAS-Outpatient protocol against standard management (Table 1) + Patients’ Satisfaction on a Visual Analog Scale (Table 2) – both studies completed at our centre.

HMR hospital in Montreal has taken the lead in developing an ERAS system that not only optimizes patients, it integrates ERAS with outpatients and succeeds in delivering an ERAS-outpatient pathway (Table 1 included).  

Reviewer 2 Report

The COVID-19 pandemic has impacted healthcare systems across the globe and the allocation of healthcare resources has been changed. The implementation of ERAS-Outpatient protocol in THA is meaningful to the adjustment of medical system. To understand the protocol more accurately, there are still several questions which need further explanations. 

Additional comments:

  1. In the section of “Patient optimization”, author mentioned the preoperative standards of HbA1c and hemoglobin, but did not demonstrate the further management plan for patients who could not be adjusted to the corresponding levels before surgery (Delayed surgery? Rehab center? Hospital admission and specialized treatment?). In addition, authors did not specify the exact time point at which it was safe for patients to stop intra articular injections before THA surgery.

  1. The delay of postoperative early mobilization in a number of the elderly patients is limited by decreased muscle strength and poor balance function, which is crucial for the ERAS-outpatient protocol after THA surgery. Authors should provide a more detailed description of the perioperative improvement in the two aspects.

  1. The ERAS-outpatient protocol might be a promising option to allow safer and better recovery after an outpatient procedure for THA patients. However, the requirements for implementation of the protocol are stringent, and not all health systems can completely meet it. How to widely practice the protocol should be more detailed.

Reviewer 3 Report

This study conducted by Sivaloganathan et al. presents a recovery protocol that can be implemented in patients with hip replacement. In general, it is an average written manuscript, but there are aspects that needs to be pointed out. The protocol itself is not clearly understandable by just reading it a plain text. When going through the text itself it feels like a review, but also like an enumeration of factors that aid total hip replacement. 

Lack of diagrams, tables and sketches make the manuscript extremely unsatisfying to read for modern science.

A major limitation of the protocol is the absence of a test group and/or a clinical application of the described factors. However, this is not the aim of the study, but could greatly improve the quality of the manuscript and protocol.

Row 33 - please consider changing the word "recuperation" with something more appropriate like "rehabilitation", "recovery";

Row 69 - consider writing Diabetes with lower-case letters;

Row 81 - consider adding a sentence describing what "at risk patients" mean;

Row 86 - do you have a reference that can sustain that statement? Please add it;

Row 87-88 - please be advised that your statement might interfere with anticoagulation regiments that certain patients receive. Provide a statement with a reference for that issue;

Row 92 - this phrase does not make sense in English - consider rephrasing;

Row 95 - what type of clorhexidine wash are you refering to? Please be more specific;

Row 96 - we know that it is contraindicated but consider to  also add a complication of razor shaving and not only mention it;

Row 103-104 - rephrase this sentence. Add a more scientific and elaborated conclusion to the entire paragraph of infection prevention;

Row 106 - THA abbreviation has not beed described in the manuscript - please add full names of abbreviations;

Row 108-109 - lets add more value to this paragraph - add a more detailed description for each of the blood loss management issues (eg - for hemostasis, tranexamic acid usage etc.);

Row 130 - please specify what type of cryotherapy - local? general? In the orthopaedic field, we all know, but for a scientific article you have to carefully mention;

Row 162 - Even though DVT is a known abbreviation, please specify what it means;

Rows 167-169 - ASA, TJA, VTE - please provide explanations for these abbreviations;

Row 178 - add a "." at the end of the sentence;

Row 209-210 - you mention that following a clear defined goal will simplify postoperative care; however, the manuscript does not provide any figure or sketch that may aid the reader when applying the protocol;

English writing and grammar is a throwback on the current state of the manuscript. A professional english edit is required.

Overall, at the current stage of the manuscript, I do not feel like it brings new insights for the topic, especially with the lack of clear guidance on how to concisely apply the ERAS clinically. However, with the addition of diagrams, flowchart or sketches and a clear explanation of step-by-step protocol, the manuscript could be improved.

Round 2

Reviewer 3 Report

Authors positively added the requested files. However, I see an image quality issue on the tables provided. Also, please add a correctly written legend to the diagram and table provided with their correct representation in text.

Author Response

Response to Reviewer 3 (Part 2)

Tables have been amended

Legends have been amended:

Table 1: Complications rates per group (Control Vs ERAS short-stay) according to the Clavien-Dindo classification [7].

Table 2: Patients’ Satisfaction on a Visual Analog Scale (0–100) [30]. Comparison of ERAS-outpatient group and the control (standard) group illustrating superior patient satisfaction scores.

Illustration 1: Flow diagram illustrating the patient pathway from the point of patient selection to the post operative period.

Correct representation in the text has been made

Line 430 (illustration 1: Flow Diagram)
